

# Mate-guarding duration is mainly influenced by the risk of sperm competition and not by female quality in a golden orb-weaver spider

Lygia A. Del Matto[1,2], Renato C. Macedo-Rego[1,3,4] and
Eduardo S. A. Santos[1,2,3,5]

[1] BECO do Departamento de Zoologia, Universidade de São Paulo, Sao Paulo, Brazil
[2] Programa de Pós-Graduação em Zoologia, Universidade de São Paulo, Sao Paulo, Brazil
[3] Programa de Pós-Graduação em Ecologia, Universidade de São Paulo, Sao Paulo, Brazil
[4] Research School of Biology, Australian National University, Canberra, Australia
[5] RH Lab, Banco Santander, São Paulo, Brazil

Corresponding author
Eduardo S. A. Santos,
e.salves@gmail.com

## ABSTRACT

Males are expected to mate with as many females as possible, but can maximize their reproductive success through strategic mating decisions. For instance, males can increase their own fitness by mating with high quality females that produce more offspring. Additionally, males can adjust mating effort based on the relative distribution of females and male competitors. To test factors that influence male mate choice, we assessed male mating decisions in the golden silk orb-weaver spider, *Trichonephila clavipes* (Nephilidae), a species in which females are polyandrous, males guard females before and after copulation occurs and large males are the most successful at guarding mates. We tested the hypothesis that males spend more time guarding high quality females that are spatially isolated, and when the risk of sperm competition is higher. We also hypothesized that this effect increases with male body size. We assessed solitary and aggregated female webs in the field and quantified female quality (*i.e.*, female body condition), male size (*i.e.*, male body size), the risk of sperm competition (*i.e.*, number of males in each female web), and mate-guarding duration (*i.e.*, number of days each male spent in each web). We found that mate-guarding behaviour is largely influenced by the presence of male competitors. In addition, male body size seems to moderately influence male guarding decisions, with larger males guarding for a longer time. Finally, female body condition and type of web (*i.e.*, solitary or aggregated) seem to play small roles in mate-guarding behaviour. As mate-guarding duration increased by 0.718 day per each additional male competitor in the web, and guarding behaviour prevents males from seeking additional mates, it seems that guarding females can be considerably costly. We conclude that failing to guard a sexual partner promotes high costs derived from sperm competition, and a male cannot recover his relative loss in fertilization success by seeking and fertilizing more females. In addition, the search for more sexual partners can be constrained by possible high costs imposed by weight loss and fights against other males, which may explain why the type of web only moderately influenced male mate choice. Following the same rationale, if high-quality females are not easy to find and/or mating with a high-quality female demands much effort, males may search females and guard them regardless of female

quality. In conclusion, the factor that most influences male mate-guarding behaviour among *T. clavipes* in the field is the risk of sperm competition.

## INTRODUCTION

The classical theory of sexual selection predicts that male reproductive success is positively influenced by the number of females a male copulates with (*Darwin, 1871*; *Bateman, 1948*). Therefore, following the Darwin–Bateman paradigm, males were initially predicted to mate with the maximum number of females they can have access to (*Dewsbury, 2005*). However, factors such as the quality of a potential mate, the difficulty in finding additional matings, or the intensity of sperm competition should all influence the expected reproductive success of males (*Scharf, Peter & Martin, 2013*). Consequently, developments in sexual selection theory lead to predictions that males should adjust their mating investment based on expected reproductive success (*Parker, 1998*; *Bonduriansky, 2001*). This strategic adjustment of male mating investment should occur especially in species in which males face high costs of mating or limited mating opportunities (*Schneider, 2014*). Thus, male reproductive success is not always maximized by copulating with as many females as possible.

Male strategic mating decisions are likely influenced by female quality (*Bonduriansky, 2001*; *Edward & Chapman, 2011*). For males, high quality females may be the females with the best body condition, given that they are more likely to survive until oviposition and provide resources for the eggs (*Rittschof, 2011*). Female body size can indicate female condition for males, and it is also a predictor of female fecundity in arthropods (*Honěk, 1993*; *Foellmer & Moya-Laraño, 2007*). Thus, copulating with large females could increase a male's reproductive success as a result of a greater likelihood of female survival until oviposition and also as a result of an increased number of viable eggs a male can fertilize. In scenarios in which female condition is variable, males can optimize their reproductive success by choosing to mate with females that are in the best condition (*Reinhold, Kurtz & Engqvist, 2002*). Thus, mating with a few high-quality females may yield a greater reproductive success than mating with many low-or average-quality females (*Bonduriansky, 2001*).

Complementarily, male strategic mating decisions may be influenced by the spatial distribution of females and male competitors. For instance, whether females are scattered or aggregated around a habitat should influence how males encounter females. This spatial distribution dictates whether males encounter females in a more simultaneous or sequential manner, and male mate choice is expected to evolve under the former condition (*Barry & Kokko, 2010*). For males that have to search for females, finding more than one viable partner can be easier when females are "clumped", as there can be "hot-spots" of reproductive opportunity (*Emlen & Oring, 1977*). However, when the costs of finding

females are low, the probability of facing male competitors and the risk of sperm competition tend to increase (*Gage, 1995*; *Kokko & Rankin, 2006*). Thus, when investigating aspects of male mate choice, one should take into account the distribution of potential mates as well as the distribution of competitors. Whenever different females can be easily found, males could optimize their reproductive success by investing in a mating strategy of searching for several mates (*Kasumovic & Jordan, 2013*). On the other hand, whenever several males compete for the same female, or females are hard to find, males could optimize their reproductive success by defending the access to the female (s) (*Seibt & Wickler, 1979*).

In the golden silk orb-weaver spider, *Trichonephila clavipes*, females can mate with multiple males (*Rittschof et al., 2012*) and males guard females both before mating, when females are immature (*Christenson & Goist, 1979*), and after mating (*Cohn, Balding & Christenson, 1988*). Females construct webs that can be physically connected to the webs built by other females or can be completely isolated (*Rypstra, 1985*). Males are able to move to different webs and sample females, which may allow them to choose the best options among potential sexual partners (*Pollo, Muniz & Santos, 2019*). Additionally, males become sexually mature sooner than females in the breeding season, thus they can mate-guard juvenile females and wait to copulate when they become sexually mature (*Christenson & Goist, 1979*). Importantly, it is known that large males have an advantage in intrasexual contests (*Rittschof, 2010*; *Constant, Valbuena & Rittschof, 2011*), assuming hub male status and guarding females more frequently than small (satellite) males (*Christenson & Goist, 1979*). Moreover, male competitive ability has been shown to influence male mate choice. Large males have a preference for large females and small males prefer small females (*Pollo, Muniz & Santos, 2019*). This assortative pattern is particularly relevant because large females are more fecund (*Rittschof, 2010*). Therefore, good competitors (*i.e.*, large males) tend to achieve higher reproductive success than small (satellite) males.

Given that males in *T. clavipes* can adopt different strategies (*i.e.*, guarding behaviour *vs.* searching behaviour) and the outcome of these strategies depend on how much females vary in quality and on how easy it is to find females, we asked: why do some *T. clavipes* males spend more time guarding a given female instead of searching for more females, while other males do not? Our hypothesis is that males spend more time guarding high quality females that are spatially isolated, when the risk of sperm competition is higher and when the guarding male is larger. Therefore, we predict that males will guard females for longer periods of time when (1) females are in solitary webs that are not physically connected to other female webs, (2) there are more satellite males in the female web, (3) females are in a better body condition, and (4) guardian males are larger. We also test predictions based on two-way interactions between these variables, as these are of biological interest. Males (especially large males) should invest more time guarding females in good body condition when in solitary webs, and when there are more satellite males.

## METHODS

### Data collection

We conducted our study in the gardens that surround the Zoology Department building at Instituto de Biociências (23.564°S, 46.729°W) on the main campus of the University of São Paulo, in São Paulo, Brazil. The garden is delimited by the Zoology Department building and by the access roads around it. *T. clavipes* individuals present in the area build their webs using trees, shrubs, lamp poles and walls as anchoring substrate. We made observations twice a day, at approximately 9 A.M. and at 2 P.M. between the 11th of February and the 23rd of May of 2015. We did not collect data (see below, for details) when there was heavy rain, because under these conditions females can eat up to half of their webs (L.A. Del Matto, 2015, personal observation), influencing the position of males, and, thus, our data collection.

We limited our data collection to those webs that were built between 0 and two m above the ground (measured from the hub of the web to the ground) in order to allow us to individually mark individuals and conduct accurate observations. During each visit to a web, we classified it as "solitary" or "aggregated". If the web did not share threads with other webs, we considered it as a solitary web. If the web shared threads with other webs, we considered it as an aggregated web. The type of web could change from day to day, with new females sharing threads with a previously solitary web or females becoming isolated because other females on the aggregation died. We identified females by combinations of colours painted in the dorsal side of their abdomens using acrylic paint (Testors). We conducted this procedure without removing the females from the web to avoid disturbances that may cause females to abandon their webs. In some aggregations, painting females' abdomen was not possible without removing threads or destroying adjacent webs. Therefore, some females were identified daily by their web position in the aggregation, as positions hardly changed from day to day (*Vollrath, 1985*; L.A. Del Matto, 2015, personal observation). Additionally, because aggregations prevented us from accessing some females without damaging webs, we could not accurately identify female age by assessing their epigynum. Nevertheless, female age (penultimate instar females *vs.* moulting females *vs.* mature females) has little influence on female attractiveness in another Brazilian population of *T. clavipes* (*Almeida & Peixoto, 2021*).

We also identified males with fine dots of acrylic paint on the dorsal side of their abdomen. Each male received a unique combination of two or three colour dots that allowed individual identification. In order to mark males when monitoring female webs, we removed the male from the web with a fine paintbrush and placed him inside a Petri dish with a small scale bar if the male was not uniquely identified before. We placed this Petri dish on top of ice and kept it inside an ice-cooler for approximately 2 min. We used this cooling procedure to make males less active in order to facilitate individual colour-marking and photographing males for posterior measuring. Each male received a unique combination of two or three colour dots that allowed individual identification. We recorded males as "guardians" if they were occupying a central position on the web and were the closest male to the female. We identified males as "satellites" if they were on

sustaining threads or on the periphery of the female web and were not the closest male to the female. We returned males by placing them individually inside a plastic cup and holding it as close as possible to the male's original position on the web. Later, with the photos we took of males, we used the software Image J (*Schneider, Rasband & Eliceiri, 2012*) to calculate male cephalothorax width (mm), our proxy for male body size.

We photographed females at intervals of approximately 10 days. All photos contained a scale bar for posterior measuring. We measured cephalothorax width (mm), abdomen width (mm), and abdomen length (mm) of females using Image J (*Schneider, Rasband & Eliceiri, 2012*) to infer body condition, our proxy for female quality, as larger females produce more offspring (*Rittschof, 2010*) and females with higher body condition are more attractive (*Rittschof, 2011*). If more than one photo of a female was taken, the photos used for measurement were the ones that were taken closer to the period a male stayed with the female. Female abdomen volume was estimated based on abdomen length and width, and assuming its shape to be equivalent to the volume of a cylinder $V = \pi \times r^2 \times h$. We used residuals from a regression between the logarithm of abdomen volume (response variable) and cephalothorax width (predictor) to calculate a body condition index. This body condition index provides an estimate that is uncorrelated with body size, and this approach is widely used in studies with spiders (*Taylor, Price & Wedell, 2014*; *Macedo-Rego et al., 2016*). Negative values represent females with smaller predicted abdomen volume given their cephalothorax width, whereas positive values represent females with larger predicted abdomen volume given their cephalothorax width.

## Data analysis

To test our predictions about the amount of time guarding males spent with females, we ran generalized linear mixed models (GLMMs) with model selection and model averaging based on AIC (*Burnham & Anderson, 2002*). The response variable of the models was time spent (in days) by a guardian male with the same female. Thus, we fitted GLMMs with a Poisson error structure and a log link function. In all models, we used male identity as a random factor to account for the repeated observations made on individual males. Moreover, to account for overdispersion, we included an observation level random effect in the model. We included the type of web (aggregated or solitary) as a categorical binary predictor. The risk of sperm competition was included as a predictor variable and was coded as the maximum number of males present on the web during the time a guarding male spent with the same female. We also included female body condition index and male body size as continuous predictors. We included two-way interactions between female condition, male body size, number of males, and type of web in the global model. We standardized all input variables with the function rescale from the package *arm* (*Gelman & Su, 2016*) to be able to directly compare effect sizes of the predictors and to allow comparison of the effects when interactions are present.

We used model averaging (*Burnham & Anderson, 2002*; *Grueber et al., 2011*), with the functions available in the package *MuMIn* (*Bartón, 2018*), to determine what were the most important predictors in our GLMM. We built a global model with all predictors and interactions based on our predictions, and then derived a set of models with all

**Table 1 Top model set (ΔAIC$_C$ < 2.0) for the number of days a male *Trichonephila clavipes* spent guarding a female (*n* = 93 observations).**

| Fixed effect predictors | AIC$_C$ | K | ΔAIC$_C$ | Weight |
|---|---|---|---|---|
| Male body size + number of males + female condition + number of males × female condition | 392.47 | 7 | 0.00 | 0.42 |
| Number of males + female condition + number of males × female condition | 392.61 | 6 | 0.14 | 0.39 |
| Male body size + number of males + female condition + number of males × female condition + male body size × female condition | 394.13 | 8 | 1.67 | 0.18 |

Notes:

All models include the random effects of the male identity and also observation level. Models are ranked by increasing order of their Akaike information criterion corrected for small sample size (AIC$_C$).

K = number of parameters, ΔAIC$_C$ = difference between the AIC$_C$ value of each model and the AIC$_C$ value of the most parsimonious model, and weight = AIC$_C$ weight of each model. The symbols + and × represent additive and interaction between variables, respectively.

combinations of explanatory variables. We defined our top model set as those models that fell within two AICC of the best model in the set. We used the natural-average method to conduct model averaging (*Burnham & Anderson, 2002*; *Grueber et al., 2011*; *Nakagawa & Freckleton, 2011*). Model averaging yields two outputs, the standardized coefficients (and their unconditional standard errors, which incorporate model-selection uncertainty) and the relative importance of each coefficient for explaining the variance in the response variable (*Grueber et al., 2011*). We present estimated parameters along with their 95% compatibility intervals (CIs) and discuss our findings interpreting the parameter point estimates, while at the same time acknowledging their uncertainty (*Wasserstein, Schirm & Lazar, 2019*). All GLMMs were built using the *lme4* package (*Bates et al., 2015*) in the R programming environment (*R Core Team, 2017*).

## RESULTS

We sampled a total of 83 males and we classified 39 of them as guardian males. We sampled a total of 31 females (22 aggregated and 14 solitary; type of web sums to 36 webs because some females changed between types). The median number of males—including guarding and satellite males—in a web was 1 (range: 1 to 5; aggregated webs: 1 [1 to 3] male; solitary webs: 1 [1 to 5] male). We re-sighted 25 guardian males in at least another web of a different female than the one he was originally observed with. Female body condition, estimated by the residual of abdomen volume and cephalothorax width, ranged from −492.14 to 516.46 (mean female body condition = −6.94, S.D. = 243.0). Male body size ranged from 1.08 to 2.68 mm (mean male body size = 1.97 mm, S.D. = 0.37 mm). The average distance between focal webs and the closest neighbour web was 9.21 ± 6.55 m S.D. for solitary webs and 10.5 ± 8.5 m S.D. for aggregated webs.

We generated a model set (including the null model) from the global model that resulted in 49 models (see https://doi.org/10.6084/m9.figshare.13522136.v6 for dataset). Including the best model, we had three models within two AICC top model set (Table 1). All the top models contained the predictors "number of males", "female condition" and the interaction between "number of males" and "female condition". Two models contained the predictor "male body size", and one model contained the interaction between "male body size" and "female condition" (Table 2).

The parameter estimate for the number of satellite males on the web was positive, indicating that the amount of time a guarding male spent with a female increased as the

Table 2 Standardized predictors, from the averaged model, of the number of days a male *Trichonephila clavipes* spent guarding a female.

| Parameter | β | SE | 95% CI |
|---|---|---|---|
| intercept | 0.895 | 0.118 | [0.660–1.130] |
| Male body size | 0.191 | 0.124 | [−0.055 to 0.438] |
| Number of males | 0.718 | 0.182 | [0.357–1.079] |
| Female condition | −0.233 | 0.199 | [−0.629 to 0.163] |
| Number of males × female condition | −1.297 | 0.353 | [−1.998 to -0.596] |
| Male body size × female condition | −0.151 | 0.176 | [−0.500 to 0.198] |

**Notes:**
Results shown are model predictors derived after averaging submodels within 2 AIC$_C$ of the best model.
β, standardized coefficient for model predictors; SE, unconditional standard error; 95% CI, 95% compatibility interval.

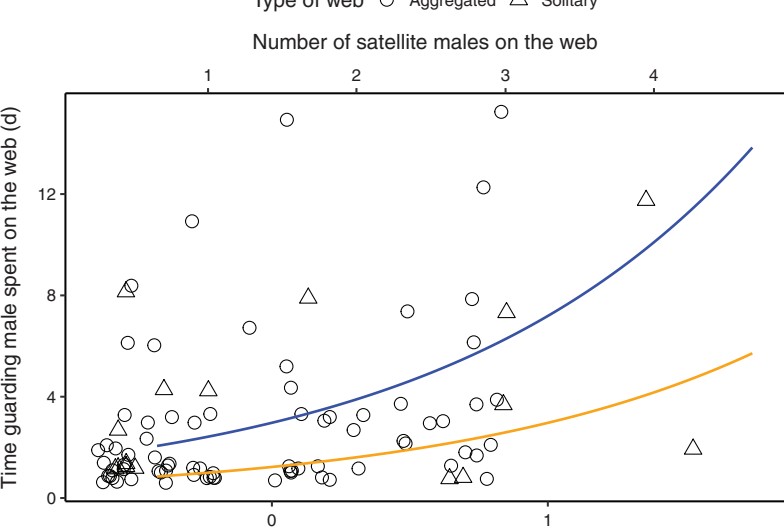

**Figure 1 Relationship between the number of males (primary x-axis: centred and scaled; secondary x-axis: original data) on the female web and the time the guarding male spent (in hours) with a female on her web.** Aggregated webs are represented by circles; solitary webs by triangles. Regression line (in blue) shown from coefficients of average model. Original points have been jittered horizontally to minimize overlap. One observation in which a male spent 32 days in a female web was removed from the plot for better visualization.

number of males on the web increased (Table 2, Fig. 1). The 95% CI for the number of males on the web ranged from 0.357 to 1.079, suggesting that our data is compatible with small to large positive effects of the presence of satellite males on the time spent guarding (Table 2). The male body size parameter was also positive, but the confidence interval overlapped zero (Table 2). The interaction between the number of satellite males on a web and female condition was negative, meaning that mate-guarding duration decreases as both the number of competitor males and female quality increase. The respective 95% CI ranged from −1.998 to −0.596 which is compatible with a small to large negative effect of this interaction term on the time spent guarding (Table 2). The female condition parameter and the interaction between male body size and female

condition were negative but presented unconditional standard error estimates (which incorporates model-selection uncertainty) that were large relative to their effect sizes. Thus, the 95% compatibility intervals for these estimates overlapped zero, suggesting that these results are most likely compatible with no important effects (Table 2).

## DISCUSSION

Here, in a population of *T. clavipes*, we showed that the number of males cohabiting a female web is the factor that, alone, best explained the variation in male mate-guarding behaviour. In other words, males seem to guard females for longer periods when the risk of sperm competition is higher, and sperm competition seems to be the factor that influences male mating decisions the most. Importantly, we also showed that the intrinsic quality of females interacts negatively with the number of males on a web, which indicates that guardian males reduce guarding time when higher quality females have more satellite males on their webs. Additionally, we showed that male body size may moderately influence male mating decisions, with larger males investing more time in guarding position, but note that the compatibility interval of this estimate overlapped zero. Regarding the interaction between male body size and female quality, we found a small reduction in mate-guarding duration among larger males as higher the female quality was, but the compatibility interval overlapped zero. Similarly, we showed that female quality alone seems to have a small negative influence on mate-guarding duration, but the compatibility interval overlapped zero, again. Finally, we found that web connectivity had little to no influence in mate-guarding duration, once none of the selected models included the type of web. Below, we discuss why the risk of sperm competition is more effective in shaping male mate-guarding behaviour than web connectivity, female quality and male size.

As the number of satellite males on a web increases, it is more likely that some of these males will manage to copulate with the female, leading to sperm competition for the fertilization of her eggs. This is a probable scenario given that it is now well-known that females mate multiply (*Gowaty, 2006*; *Taylor, Price & Wedell, 2014*), with several examples among spiders (*e.g.*, *Prokop & Maxwell, 2009*; *González, Costa & Peretti, 2019*), including *T. clavipes* (*Rittschof et al., 2012*). A male may avoid the risk of sperm competition by guarding a female and, therefore, preventing competitors from mating with her, as widely documented for spiders (*e.g.*, *Herberstein et al., 2005*; *Elias et al., 2014*). Moreover, the higher the number of competitors, the higher should be the effort employed by guarding males. Accordingly, in our study, mate-guarding time increased 0.718 day per each additional satellite male on the web, which corresponds to 3.42% to 5.13% of a male's life as an adult (*Brown, 1985*). Therefore, if a male is guarding a female that attracted several satellite males, this guardian male tends to spend a great part of his adult life guarding one single female, instead of increasing his reproductive success by mating with a second female. In addition, despite the occurrence of sperm limitation in *T. clavipes* (*Michalik & Rittschof, 2011*), males might not transfer all their sperm to a single female (*Christenson & Cohen, 1988*; *Cohn, 1990*; *Rittschof, 2011*; *Schneider et al., 2011*), which allows them to mate again. Consequently, losing the

opportunity to mate with a second female can impose a high fitness cost. Given this, we can ask: if guarding females can be so costly, why do *T. clavipes* males guard females? Probably because the fitness costs of seeking additional mates and facing sperm competition are higher than the costs of mate-guarding behaviour, and males have a higher fitness return when reducing their lifetime mating success and guarding their current sexual partners.

Contrary to the number of satellite males present, whether a web is clustered or isolated seems to be a less important factor in determining the amount of time that a male spends guarding a female in *T. clavipes*. At first glance, one could consider that both the number of males and the type of web inform the risk of sperm competition and should influence mate-guarding duration. However, we know that the number of satellite males on a web is not related (see Supplemental Material) to the type of web, and that males in *T. clavipes* arrive at aggregated and solitary webs in equal frequencies (*Meraz, Hénaut & Elgar, 2012*). Therefore, the type of web is not a good proxy for the risk of sperm competition and any influence of web type on mate-guarding is probably explained by other variables. However, guardian males on aggregated webs still have a clear path to at least one other female in case they decide to leave their current partner, while guardian males on solitary webs are not able to access other female webs easily, and would be less keen on leaving their web. Surprisingly, the effect of web connectivity alone on mate-guarding duration is small to non-existent, which may indicate that, despite the connectivity provided by aggregated webs, leaving a given web and mating multiply brings high costs for males, regardless of web type. These costs of moving among webs and searching for additional mates might include weight loss (*Meraz, Hénaut & Elgar, 2012*) and fighting with competitors (*Rittschof, 2010*). These may explain why the type of web has little to no influence on mate-guarding behaviour in the studied spider. Another possibility is that connectivity between webs is not as important as the distances between them. As the distances are not affected by web connectivity in the studied population, this can explain why mate-guarding behaviour was little or not influenced by web type.

The possible costs derived from looking for high-quality mates and fighting with many competitors may also explain why males did not spend more time guarding high-quality females. If the search for females incurs severe survival costs for males (see *Kasumovic et al., 2007*; *Berger-Tal & Lubin, 2011*), the distribution of males among female webs may be little influenced by female quality, especially if high-quality females are not easy to find. Given this and that looking for a high-quality female probably means facing extreme intrasexual competition, males might avoid leaving a current sexual partner, regardless of its quality. However, one should not conclude from our study that males in *T. clavipes* never express mate choice. Recent papers show that male mate choice is more common in nature than previously expected (*Edward & Chapman, 2011*), and field experiments have shown that large *T. clavipes* males prefer large females when given the option to guard one of two females of different body size (*Pollo, Muniz & Santos, 2019*). In addition, we highlight that our study is restricted to mate-guarding behaviour. Therefore, it is possible that male mate choice occurs in contexts other than

mate-guarding, such as pre-mating courtship (*e.g.*, which female should the male court?) or post-mating competition (*e.g.*, how much sperm should the male transfer to a female?). In fact, *Rittschof (2011)* showed that *T. clavipes* males transfer more sperm to females that are close to oviposition. Interestingly, females close to oviposition are the ones that will have less opportunity to mate with additional mates, which means that their sexual partners tend to face lower sperm competition. Therefore, because we demonstrate that mate-guarding behaviour is mainly determined by the density of male competitors, our results and the results provided by *Rittschof (2011)* reinforce the idea that male reproductive effort in *T. clavipes* is mostly influenced by the risk of sperm competition.

As mate-guarding duration is mainly influenced by the risk of sperm competition and female quality has little to no influence, it would be reasonable to expect increasing mate-guarding duration in response to concomitant increases in female quality and sperm competition risk. However, we found the opposite. Why would a male reduce his guarding effort when mate-guarding is more important (*i.e.*, there are more competitors on the web) and the potential fitness benefits are higher (*i.e.*, high-quality females are more fecund, *Rittschof, 2010*)? Two non-exclusive hypotheses can explain this result. First, regarding the number of males on a web, there may be a guarding threshold, above which guardian males cease mate-guarding and leave the web (analogously, across animals, aggressive behaviour increases as the OSR becomes more male-biased and reduces after reaching a threshold; *Weir, Grant & Hutchings, 2011*). Such hypothetical threshold may be easily reached on webs of high-quality females, as they are more attractive to large males (*Pollo, Muniz & Santos, 2019*), and large males are the best competitors when it comes to fighting for the guardian position (*Constant, Valbuena & Rittschof, 2011*). Second, it is possible that satellite males behave differently according to female quality, being more aggressive towards the guardian male when competing for a high-quality female (increased male aggressive behaviour due to high female quality has been demonstrated for other taxa; *Weiss & Dubin, 2018*; *Kapranas et al., 2020*), which would make guarding effort more costly and would reduce guarding duration. In addition, as high-quality females attract more good competitors (*i.e.*, large males; *Pollo, Muniz & Santos, 2019*), it is possible that satellite males become even more aggressive and/or bold due to increased pre-mating competition, leading to an even lower mate-guarding duration.

Similarly to the interaction between the number of competitor males and female quality, the interaction between male body size and female quality is negatively correlated with mate-guarding duration. However, the influence of this interaction is much lower and the compatibility interval overlaps zero. Therefore, from a guardian's perspective, it seems that self-evaluation plays a less important role than evaluating external factors, such as quantity and quality of conspecifics. While male body size alone has a moderate positive influence on mate-guarding duration, indicating that self-evaluation has some relevance and corroborating the evidence that larger males are better fighters (*Constant, Valbuena & Rittschof, 2011*), the influence of sperm competition risk alone is more than three times higher, reinforcing the idea that external factors are the ones that best modulate mate-guarding duration. More specifically, because the number of male

competitors showed to be the most important factor influencing mate-guarding, our results provide additional evidence that post-mating competition plays an important role in determining male reproductive investment in animals (*Parker & Pizzari, 2010*). In this sense, as guarding behaviour in *T. clavipes* can occur before (*Christenson & Goist, 1979*) and after (*Cohn, Balding & Christenson, 1988*) mating takes place and both forms of guarding behaviour influence the sperm competition faced by males, future experimental studies should assess whether the risk of sperm competition influences pre-and post-mating guarding effort in a similar manner. Additionally, as pre-and post-ejaculatory investments may be traded-off (*Parker, 2020*), future studies should also evaluate how investment in mate-guarding and investment in ejaculate are related in *T. clavipes* and other species. Complementarily, given the relevance of post-mating events in male decisions and that spiders are good systems to study cryptic female choice (*Eberhard, 2004*), future studies should address how mate-guarding duration influence the way females use the sperm of each sexual partner in *T. clavipes* and other spider species.

### Funding
This study was financed by Coordenação de Aperfeiçoamento de Pessoal de Nível Superior-Brasil (CAPES)-Finance Code 001-Programa de Excelência Acadêmica (Proex) (Masters scholarship: Lygia A. Del Matto; PhD scholarship: Renato C. Macedo–Rego; article processing charge: Lygia A. Del Matto, Renato C. Macedo–Rego, and Eduardo S. A. Santos). Lygia A. Del Mattoalso received an undergraduate research scholarship from the Universidade de São Paulo for the duration of field research. Renato C. Macedo-Rego also received a grant from Programa de Doutorado-sanduíche no Exterior (PDSE/CAPES; Edital n° 47/2017; Processo 88881.189156/2018-01). The funders had no role in study design, data collection and analysis, decision to publish, or preparation of the manuscript.

### Grant Disclosures
The following grant information was disclosed by the authors:
Coordenação de Aperfeiçoamento de Pessoal de Nível Superior-Brasil. Macedo–Rego, and Eduardo S. A. Santos. Universidade de São Paulo. Programa de Doutorado–sanduíche: 88881.189156/2018-01.

### Competing Interests
The authors declare that they have no competing interests.

### Author Contributions
- Lygia A. Del Matto conceived and designed the experiments, performed the experiments, authored or reviewed drafts of the paper, and approved the final draft.
- Renato C. Macedo–Rego analyzed the data, authored or reviewed drafts of the paper, and approved the final draft.
- Eduardo S. A. Santos conceived and designed the experiments, analyzed the data, prepared figures and/or tables, authored or reviewed drafts of the paper, and approved the final draft.

## Data Availability

The data is available at Figshare: Santos, Eduardo; Del Matto, Lygia; C. Macedo–Rego, Renato (2021): Dataset and analysis script of manuscript "Mate-guarding duration is mainly influenced by the risk of sperm competition and not by female quality in a golden orb-weaver spider". figshare. Dataset. https://doi.org/10.6084/m9.figshare.13522136.v6.

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
