# Peer review of "Mate-guarding duration is mainly influenced by the risk of sperm competition and not by female quality in a golden orb-weaver spider"

_PeerJ, doi:10.7717/peerj.12310_

## Round 0.1 · original submission · Major Revisions

Thank you for your paper. It was reviewed by three highly-qualified reviewers. While the reviewers generally agree that the data set you have produced is an important contribution to the field, there are some minor (and major) issues that need to be addressed prior to publication. Both reviewers 1 and 2 had some issues with data availability and variable naming. All reviewers had comments on how hypotheses were tested statistically. There were some concerns about how reliable the variables selected (e.g. body size, see reviewer comments) are as indicators. Reviewers 2 and 3 felt that referencing could be improved and expanded, especially when working with such a well-studied spider. Overall, I agree with the reviewers and given the breadth of comments and the major issues raised by one reviewer, I think a Major Revision is appropriate. I see some potential for the authors to address the major issues with analysis by, perhaps re-analyzing their data to account for some of these issues and increase their referencing.

Please specifically address each of the issues raised by the reviewers and resubmit a revised manuscript at your convenience. Thanks again for your contribution.

Reviewer 1 ·

Basic reporting

The female quality dataset, "nephila-female-morpho.csv", which is required to run the provided R code is not available on figshare, so this needs to be uploaded before publication.

Experimental design

Good! Nice field data.

Validity of the findings

Good.

Additional comments

This is a very interesting study investigating how the spatial distribution of females, female body condition, and risk of sperm competition affect male mate guarding effort in a nephilid spider. This kind of field study is very valuable for understanding male mate choice and how ecological context shapes mating systems. The paper is well written and the framing is appropriate, and I have only minor suggestions for improvement in terms of interpreting and displaying the results, and clarifying the predictions and choice of parameters included in the statistical model.

Is female body condition correlated with fecundity or fertility or other fitness-related variables in T. clavipes? In the abstract (lines 25-27) the authors hypothesize that mate guarding will be related to female quality, and the second paragraph of the introduction is related to this idea, so it would be helpful to indicate whether there is data showing that condition is a good measure of female quality for this species (or relatives).

In the final paragraph of the introduction the authors make predictions about how three variables (female spatial distribution, number of competitors on the female’s web, and female body condition) will affect the duration of male mate guarding (a measure of mating effort). Interactions (lines 165-159) between spatial distribution and number of males, and between spatial distribution and female condition are explicitly included in the models considered in the analysis, but not female condition and number of males. Why not include all 2-way interactions (or why expect any interactions)? If the authors have specific predictions about how these predictor variables should interact to affect male guarding effort, these should be spelled out in the introduction. Although the expected effects of each variable alone are very clear, it is not obvious to me whether or how they should be expected to interact.

Do you have any data on the actual distances between webs? Like a nearest neighbour distance? If so, it would be useful to report an average nearest neighbour distance for isolated vs. clustered webs. It seems like this population is quite dense and since the criterion for isolation is only that webs do not share silk threads, it may be that the actual distances involved tend to be fairly similar whether or not the web is classified as isolated. If so, this could be driving the apparent lack of importance of female spatial distribution in the model reported here.

Line 197/line 226: Males spent on average 2.5 days guarding females, and guarding duration increased by 0.71 days for each additional competitor on a web. What is the biological significance of this? Do the authors have any data on how guarding effort affects male fitness outcomes in terms of the difference an additional day or two of guarding makes? Are females less likely to be visited/detected by males within days after mating, for instance because they cease pheromone production? Or are females more likely to use the sperm of males that guard them for longer, or to reject subsequent males after being guarded? Some discussion of what is (or is not) known about this would help the reader understand the potential significance of mate guarding in this system.

As a reader I would appreciate graphs or tables with summary statistics for the actual number of days males spent guarding isolated vs. clustered females and the relationship between female quality and guarding duration in addition to the existing graph (fig 1) showing the relationship between number of competitors and guarding duration. This would be particularly relevant for the web type variable since it does end up in the final model, but all of the data should be reported whether or not it has a significant effect on guarding effort. It would also be preferable to show the raw data rather than scaled and centred data in figure 1, if possible, so the reader can see the actual range of male competitors in the field.

(line 243) “Assuming that guarding males in aggregated webs are more exposed to competitors” are they? Does the data from this study indicate that the number of competitors is related to web type? Later (line 247-248) the authors state that males arrive at aggregated and solitary webs at equal rates, so its not clear where this assumption comes from or whether it is justified.

(line 246) “the interaction between these factors seems to explain variation in male mate guarding behaviour” this is a bit confusing to me, because based on the results, there was no statistical interaction between web type and number of males.

Line 251: “surprisingly, the effect of web connectivity alone is moderate to nonexistent” could this be because all webs are relatively close together, whether or not they are connected by silk threads?

Line 270 “males prefer large females” can the authors expand on how preference was measured here? Are males more likely to visit large females when offered a (simultaneous) choice between females? Can you elaborate on how mate guarding fits in to mate choice more generally in this species. For instance, do males choose who to visit based on female quality, when given a choice, but once they are on a web the effort they put into mate guarding may depend more on risk of sperm competition? This could be discussed in a framework of episodes of selection.

Specific comments

Line 71: replace “various” with “more”

Line 186: should be “type of web sums to 36”

Line 267: should be “recent papers show”

Line 294: should be “systems to study”

Line 34: instead of “considerably” state the actual increase (0.71 days of guarding per competitor)

Reviewer 2 ·

Basic reporting

The writing is good, but the relevant literature is not sufficiently considered

Experimental design

The data are observational and the analyses correlational. As such it does not fill a knowledge gap. However, the dataset is of very high value and field studies over such a long time period are highly desired.

Validity of the findings

As I state in my comments to the authors, the analyses stays far below the potential that the data may provide. As it is, the conclusion are not well justified and do not add significantly to existing knowledge

Additional comments

The authors report results of a field study on Trichonephila clavipes where individually marked males and females were closely observed over a period of almost 4 months. Goal of the study was to explain causes of variation in male mate guarding duration and the main predictors chosen were the number of males on the same web, the size of the female and whether the web was isolated or aggregated.
Data were analysed using generalised mixed models that contained the above predictors plus random factors that account for the repeated measures design. They revealed that the number of males on the web explained a significant part of the variation in male web tenure. The authors conclude that sperm competition is the main factor that drives mate guarding decisions in males.
While field studies on spiders are still rare and highly welcome, I fear that the present study failed to utilize its potential. Twice daily observations across a very long period of a spider`s life are extremely rare and promise very interesting insights into the mating dynamics of a local population. As it is, analyses and results remain rather superficial and do unfortunately not provide novel insights into the context of male decisions making and strategic investment in mate guarding.
T. clavipes is among the best studied orb-web spiders but the authors chose to ignore much of the knowledge concerning T. clavipes and the other species from the same genus. This is a pity because this knowledge would have been very useful to guide the study design and the selection of variables for analysis.
To make my point clearer, I will briefly sketch a few examples:
Firstly, female body size was selected to estimate female quality but this variable did not explain much of the variation in guarding time. I would argue that it is not surprising that female size did not predict male behaviour well, since it is known that it is rather female age since maturity and her mating status that are relevant factors in this respect (see Rittschof papers). Unfortunately, there is no mention of female adult age and it is not even reported whether all females were mature or moulted during the observation period, which something that is rather easy to score. The observation period covered many weeks and next to differences in maturation and mating status, most females probably produced egg-sacs during that time. The abdomen measurements should show this since Nephilid eggsacs are very large and the abdomen shrinks considerably. Hence, condition as measured can mean different things, a female may have not eaten enough or produced eggs.
Secondly, the authors report In the methods that males were captured, marked and measured before they were released back. This is nice because males exercise size dependent mating strategies. However, male size is for unknown reasons not included in the analyses.
Thirdly, it is well established that males are sperm limited and it appears that they use up all their sperm when they mate with a virgin. However, they do not do that but retain some sperm if they mate with a mated female. Mating status strongly affects the value of the female for a given male. A male that mates with a virgin female should invest maximally in defending her against rivals since he can only increase his reproductive success by reducing the risk of sperm competition. In contrast, a male that mated with an already mated female retains potential for different options. Virgin females attract more males than mated females which could explain the correlation between male numbers and guarding duration of the hub male.
Surely, the number of males on the web might be a highly relevant parameter for male decision making but whether the best strategy is to stay or to leave is influenced by numerous other factors that have already been shown to be important. The data in this paper are all left and right censored, since nothing is known about the past of the females and males that were observed.
In conclusion, I regret to say that no reliable conclusion on the causes of male guarding behaviour can be drawn from the analyses. I very much hope that more data were collected and are available for being added, perhaps even moults were noted. Changes in female states could be better interpreted if moults were known. The dataset is great, and I would like to encourage the authors to dig a bitter deeper.

Reviewer 3 ·

Basic reporting

The article has a standard structure consisting of Introduction, Methods, Results and Discussion. It also includes two tables and one figure. The figure might be more informative with unscaled and uncentered x axis. The dataset with the raw data and the R code is provided but the names of the variables in the dataset are in Portuguese. The authors should translate it into English. The article is written in clear English, however, as a non-native speaker I cannot properly asses the standard of the language.
The background is concise with sufficient references provided. The only thing which may not be sufficiently introduced is the mate guarding in the study species – authors mention that males may guard females before maturity but it is not totally clear whether they guard only juvenile females or whether they continue guarding them also after the final moult. If yes, do the males guard females also after mating? Are the authors able to distinguish the pre- or the post- maturity/copulatory guarding? Do they focus on a particular type or on mate guarding in general? It could be mentioned briefly in the Introduction and more detailed information on that matter could be provided also in the Methods. If it is impossible to determine, potential consequences of different mate guarding should be discussed in the Discussion.

Experimental design

The authors define several hypotheses which they experimentally test in natural conditions. They provide data relevant for these hypotheses; nonetheless, they could provide more information to better understand the lifestyle and mating tactics of the study species as mentioned above and below. Also, they ask ‘why some males spend more time with the female instead of searching for more females’ (Line 98 - 100). Since they focus on time spend on mate guarding, not really on male searching behaviour or on male individual differences, I would recommend to rephrase this question.
The authors collected the males to mark them (Line 130 – 140). Do they put them in the same position on females web after the handling? Did males stay in the original position?
The type of mate guarding (before or after mating? of juvenile or adult females?) observed within this study should be clarified in Methods (or Introduction). Authors should provide information whether all observed females were adult or not and how did they determine it. Was there any way to know female status (virgin/mated)?
Also, an additional information on web connectivity (Line 116 - 129) should be added to clarify how isolated an isolated web could be – what is the minimum/maximum or average distance from other webs of webs marked as isolated? Could it be that, although not directly connected, ‘isolated’ web can be very close to other web(s) and therefore easily reachable by males?
I wonder whether the change in web connectivity could bias the data (Line 185 – 186). If connectivity is relevant, how quickly would males react to such change? Have you tried to analyse only the data including females whose web connectivity did not change?

The sentence in Line 160 – 161 is not entirely clear, please, clarify.
Why was the third possible interaction (no. of males and female condition; Line 165 - 167) not included in the model?
Can you add information on how many males visited (range; Line 188-190) and whether they remained in the same ‘role’ (satellites or guardians) if they changed the web?

Validity of the findings

The results of the study are clearly summarized and further discussed in the Discussion. You can improve it by citing more studies on mate guarding (not only on spiders) and comparing your findings with findings from these studies. Otherwise, I find the Discussion clear, the authors avoid unjustified speculations and they comment also on the negative results.

Additional comments

In general, the study is well designed and I appreciate that the authors conducted it in the field and tried to minimaze the disturbance of the spiders. Specific comments and suggestion for improvement can be found above.

---

## Round 0.2 · Minor Revisions

I thank you for your careful revision. You have handled major issues during the previous review and carefully described your findings throughout. The reviewers both agree. There are some minor issues raised by both reviewers. Reviewer 1 especially has pointed out some challenges with the paper's code and has requested changes to the discussion and interpretation of results. There are some minor grammatical changes too that you can clean up.

I look forward to your next revised manuscript.

Reviewer 1 ·

Basic reporting

Good, but a final edit for English grammar and clarity should be done on the final text.

Raw data and code shared but see comments about checking these below.

Experimental design

Good.

Validity of the findings

Good, except please see comment below about the calculation of the body condition index.

Additional comments

As before, I think that this is an interesting and valuable contribution. I thank the authors for their efforts to address my comments and those of the other reviewers. I think that the paper has improved, but I have some additional comments that I hope the authors will address.

Results:
The interaction between female condition and number of males needs to be interpreted and spelled out in the results. In the discussion, the authors state that this interaction reveals that “guardian males reduce the guarding time when higher quality females have more satellite males on their webs.” A visualization of this would be very helpful to include in the results, e.g. a plot of the type shown in Figure 1, but with female condition on the x-axis and separate points/curves for the different numbers of satellite male (0, 1, and 2).

Similarly please explain what the interaction between male size and female quality actually means. I think it is that guarding time increases with female quality, but only for small males, but this needs to be explicitly stated.

Discussion

Line 242: I am not sure it is appropriate to conclude that the main effect of the number of males cohabiting in the web is driving mate-guarding behaviour given that there is apparently an interaction between this variable and female condition. Indeed, in lines 251-253 the authors state “We also showed that the intrinsic quality of the female interacts with the number of males on a web in a negative way, which indicates that guardian males reduce the guarding time when higher quality females have more satellite males on their webs.” I think that the involvement of male number in this interaction needs to be spelled out earlier.

Lines 279-297: Thank you for adding information about nearest neighbour distances. Please include in the discussion section some discussion of the fact that the actual distances between isolated and aggregated webs do not really differ (as reported in the results, ~9 m vs. ~10 m from the nearest neighbour)—surely this also helps to explain why there does not seem to be an effect of web type. Male spiders can locomote using rapelling in addition to traversing existing silk lines between webs, so it is not obvious to me that the difference between isolated and aggregated webs in this study is really meaningful in terms of the risk of leaving to search for another female.

Analyses:
The regression model used to calculate the residuals for the female body condition index may not be appropriate, since there is not really linear relationship between abdomen volume and cephalothorax width (spider mass and weight are often log-linearly related). Using the log of the abdomen volume in the regression seems to help alleviate this problem. Re-calculating the residuals using the log volume and including this updated index in the global model does not seem to change the estimates for the fixed effects much, but I suggest re-running the analyses with updated residuals.

Please include a pdf of the code output (created using eg knitr) so that the reader can look at the analyses without having to run the code themselves (or can see how the code output should look if they are trying to run it and having trouble)

There are some issues with running the code as provided, please double check that it runs correctly with all the uploaded files on figshare. Some details:

Need to add “sep = ;” in the read.csv commands

Merging the male size data requires editing the code to account for the name of the male ID column being named “MaleID” in one dataset and male.id in the other

This issue with male.id vs MaleID comes up again in the code for resighted males and the global model etc

Minor comments:

In the abstract and Intro the authors use “golden-web spider”. Please change to the “official” common name, which (according the American Arachnological Society) is “golden silk orbweaver. ”

Line 25: change to “large males are most successful at guarding mates”

Line 33: suggest changing “mating decisions” to “guarding decisions” or “movement decisions” since the authors don’t actually measure mating decisions

Line 35-37: not clear to me that the data here indicate that males benefit from guarding, just that there is a cost…

Lines 110-11: revise to “Our hypothesis is …when the risk of sperm competition is higher and when the guarding male is larger.”

Line 154-55: delete sentence, it is a repetition of the same information already given in lines 148-49

Line 165: change to “proxy for female quality”

Please also double check grammar throughout.

Reviewer 2 ·

Basic reporting

The paper is well written throughout. General and species specific references have been added.

Experimental design

Data collection was thorough and analyses appropriate. Methods are well described

Validity of the findings

the authors carefully considered my critisism. They added male size to the analyses and explained how field work posed constrains on how well - for example - female mating status could be assessed. I understand the decision to minimise disturbance of the animals at the cost of detailed assessment of potentially relevant factors. The limitations of field work are clear to most of us but they have to be admitted and addressed. Generally, descriptive data and correlational analyses do not allow distinguishing cause and effect. Therefore, the danger of drawing incorrect conclusions from observational data alone needs to be discussed in my view (an additional problem is that many observational studies have been published including some on T. clavipes which might then get cited as providing causal evidence).
I appreciate that arguments for the use of female body size as proxy have been added and that the discussion was revised.
I also acknowledge the careful wording throughout the discussion and the frequent use of "may". While I am still not fully convinced that one can confidently conclude that the "perceived risk of sperm competition" explains male decisions to mate guard or not, I am happy with the careful discussion.
As stated before, I like the study and the data set very much.

Additional comments

there are several small grammatical errors in the revised parts and I suggest minor revision to correct them.

---

## Round 0.3 · accepted · Accept

Thank you for your revision. I have read through your changes and rebuttal and see that you have handled the reviewer's requests adequately. Reviewer 1 has agreed on this front.

I am happy to accept your revised manuscript.

[Reviewer 1 ·

Basic reporting

no comment

Experimental design

no comment

Validity of the findings

no comment

Additional comments

I thank the authors for their careful revisions and responses to my comments and those of the other reviewers!